

# Nitrous Acid Budgets in Coastal Atmosphere: Insights into the Absence of a Daytime Marine Source

Xuelian Zhong[1], Hengqing Shen[1*], Min Zhao[1], Ji Zhang[1], Yue Sun[1], Yuhong Liu[1], Yingnan Zhang[1], Ye Shan[1], Hongyong Li[1], Jiangshan Mu[1], Yu Yang[1], Yanqiu Nie[1], Jinghao Tang[2], Can Dong[1], Xinfeng Wang[1], Yujiao Zhu[1], Mingzhi Guo[2], Wenxing Wang[1], and Likun Xue[1*]

[1]Environment Research Institute, Shandong University, Qingdao, Shandong, 266237, China

[2]Collage of Mechanics and Materials, Hohai University, Nanjing, Jiangsu, 210098, China

*Correspondence to* Hengqing Shen (hqshen@sdu.edu.cn) and Likun Xue (xuelikun@sdu.edu.cn)

**Abstract.**

Nitrous acid (HONO), a vital precursor of atmospheric hydroxyl radicals (OH), has been extensively investigated to understand its characteristics and formation mechanisms. However, discerning fundamental mechanisms across diverse environments remains challenging. This study utilizes measurements from Mount Lao, a coastal mountain in eastern China, and an observation-based chemical box model to examine HONO budgets and their subsequent impacts on atmospheric oxidizing capacity. The model incorporates additional HONO sources, including direct emissions, heterogeneous conversions of $NO_2$ on aerosol and ground surfaces, and particulate nitrate photolysis. The observed mean HONO concentration was $0.46 \pm 0.37$ ppbv. The updated model well reproduced daytime HONO concentrations during dust and photochemical pollution events. During dust events, daytime HONO formation was dominated by photo-enhanced heterogeneous reactions of $NO_2$ on aerosol surfaces (>50%), whereas particulate nitrate photolysis (34%) prevails during photochemical pollution events. Nevertheless, the model uncovers a significant unidentified marine HONO source in the "sea case", with its HONO production rate reaching up to 0.70 ppbv h$^{-1}$ at noon. Without considering this unidentified source, an extraordinarily high photolysis coefficient of nitrate and/or heterogeneous uptake coefficient of $NO_2$ would be required to match observed HONO concentrations. This missing HONO source affected the peak $O_3$ production rate and OH radical concentration by 36% and 28%, respectively. Given the limited HONO observations data in coastal and marine settings, the unidentified HONO source may cause an underestimation of the atmosphere's oxidizing capacity. This study highlights the necessity for further investigation of the role of HONO in atmospheric chemistry in coastal and marine environments.



## 1 Introduction

Atmospheric nitrous acid (HONO) servers as a pivotal precursor of hydroxyl radicals (OH) (Alicke et al., 2003), accounting for up to 60% of daytime OH radicals (Kleffmann et al., 2005; Czader et al., 2012). Thus, HONO establishes itself as a critical source of OH radical source in both urban and rural environments, surpassing the contribution from ozone ($O_3$) photolysis (Elshorbany et al., 2012; Acker et al., 2006; Gu et al., 2022a). Consequently, HONO substantially influences the formation of secondary pollutants, including secondary aerosols and $O_3$, exerting a considerable effect on air quality and climate change (Xing et al., 2019; Yang et al., 2021b).

Recent studies pinpoint four primary sources of atmospheric HONO: (a) Direct emissions from traffic (Liao et al., 2021), biomass burning (Nie et al., 2015; Theys et al., 2020), and soil (Su et al., 2011). (b) Homogeneous reaction of NO + OH, which is generally regarded as a significant process in polluted urban areas during daytime when NO and OH concentrations are relatively high (Gu et al., 2022a). (c) Heterogeneous reactions of $NO_2$ on various surfaces, such as mineral dust (Underwood et al., 2001), soil (Kebede et al., 2016), and aqueous surfaces (Wojtal et al., 2011). The uptake coefficient of $NO_2$, $\gamma(NO_2)$, on these surfaces remains uncertain and is subject to varying factors, sparking debates regarding the importance of the heterogeneous conversion of $NO_2$ (Xue et al., 2022; Broske et al., 2003). (d) Photolysis of adsorbed nitric acid ($HNO_3$) and particulate nitrate ($pNO_3^-$), crucial contributors to daytime HONO formation (Gen et al., 2022; Ye et al., 2017), particularly in clean environments (Zhou et al., 2011; Ye et al., 2016a). However, HONO formation mechanisms in different environments remain contentious and require more detailed model evaluations (Jiang et al., 2022).

Over recent decades, the missing sources of daytime HONO have been extensively studied across diverse environments (Kleffmann, 2007; Jiang et al., 2022; Lee et al., 2016). However, our limited understanding of these unidentified HONO sources has hindered accurate assessments of atmospheric free radicals and oxidizing capacity (Tang et al., 2015). In areas with high concentrations of $NO_2$ and particulate matter, missing sources are often ascribed to the photolytic enhancement of heterogeneous $NO_2$ reactions (Su et al., 2008; Czader et al., 2012; Lee et al., 2016; Tong et al., 2016). Conversely, in remote areas, nitrate photolysis or soil emissions are perceived as significant contributors to daytime HONO sources (Cui et al., 2019; Ye et al., 2016a; Su et al., 2011). In polluted mountainous areas, the vertical transport of air masses may also contribute to observed daytime HONO concentrations (Jiang et al., 2020; Xue et al., 2022). During dust storms, the particle surface area increases sharply, potentially enhancing the heterogeneous reaction of $NO_2$, yet the evaluations of dust impacts on daytime HONO are scarce (Wang, 2003). Overall, most existing HONO source studies lack





quantitative assessments based on models and fail to provide comparative analyses across different environmental scenarios.

Furthermore, recent observations of HONO in coastal and marine regions indicate the existence of marine HONO sources (Jiang et al., 2022; Crilley et al., 2021; Ye et al., 2016a; Yang et al., 2021a). The observed accelerated $NO_2$-to-HONO conversion in marine air masses suggests that air-marine interactions enhance HONO production (Zha et al., 2014; Yang et al., 2021a). However, the heterogeneous conversion of $NO_2$ on vast air/water interface, a potential source of marine HONO, remains uncertain (Wojtal et al., 2011; Zhu et al., 2022; Yu et al., 2021). Nitrate Photolysis is believed to contribute to marine HONO sources (Ye et al., 2016a; Andersen et al., 2023), but significant controversy persists (Romer et al., 2018; Shi et al., 2021). The specific influencing factors remain unclear (Zhang et al., 2020; Andersen et al., 2023), with some studies suggesting other factors may be responsible (Yang et al., 2021a; Wojtal et al., 2011). However, most existing studies still rely on steady-state analysis, and there is a lack of quantitative research determining if current HONO mechanisms can adequately explain observed marine daytime HONO concentrations.

Mount Lao, located on the eastern coast of Qingdao, China, experiences influences from various air masses from the continent and the ocean. During the spring of 2021 (27 April–19 May), when dust and $O_3$ pollution occurred frequently, we conducted measurements on Mount Lao to explore the daytime HONO budgets in the coastal atmosphere. Utilizing the latest HONO formation mechanisms in the box model, we found that the existing parameters adequately accounted for the HONO sources during both dust and photochemical pollution periods. However, we identified a significant discrepancy between the simulated and observed HONO in the "sea case". This discrepancy suggests that a substantial daytime source of marine-derived HONO is absent from the current chemical mechanisms. To compensate for this missing source, either an unprecedentedly large enhancement factor (EF) of nitrate photolysis or a heterogeneous uptake coefficient of $NO_2$ would be necessary if attributed solely to these known HONO sources.

## 2 Methods

### 2.1 Field measurements

Field measurements were conducted on the southeast coast of Mount Lao (36.15°N, 120.68°E, 166 m above sea level) in Qingdao (Figure 1), approximately 1 km away from the Yellow Sea. The geographical location and elevation of Mount Lao make it an optimal location for examining the contrasts between marine and continental air masses and the chemical processes within the marine boundary layer. The relatively pristine condition of the area, coupled with minimal levels of



anthropogenic activities such as industrial emissions, establish Mount Lao as a representative of a
clean environment. The field campaign was carried out during the spring of 2021 (27 April–19 May
2021), a period when the air quality of Qingdao is often affected by dust storms from Mongolia and
northwestern China, as well as by $O_3$ pollution. Consequently, the site at Mount Lao provides an
opportune platform for investigating the fundamental formation mechanisms of HONO under diverse
environmental conditions.

HONO was quantified using a water-based long-path absorption photometer (WLPAP, Beijing
Zhichen Technology Co., Ltd, China). Ambient HONO was absorbed by deionized water alone, after
which it reacted with a reagent comprising 3.44 g of sulfanilamide and 0.2 g of N-(1-naphthyl)-
ethylenediamine-dihydrochloride (NED) in 10 liters of deionized water, leading to the formation of an
azo dye. Two channels were employed to extract HONO and interfering gases, respectively. The
absorbance of the azo dye was measured using a fiber optic spectrometer (USB 4000, Ocean Optics,
USA) at both the measurement wavelength (550 nm) and the reference wavelength (580 nm). Regular
automatic zero measurements using ultrapure nitrogen were conducted every two days to correct for
baseline drift. The detection limit and detection ranges were 2 pptv and 5 pptv–2 ppmv, respectively.

A suite of commercial online analyzers monitored the concentrations of $NO_x$, $O_3$, $SO_2$, and CO
(42i, 49i, 43i, and 48i, respectively, Thermo Fisher Scientific Inc, USA). $PM_{2.5}$ was measured using a
hybrid nephelometric/radiometric particulate mass monitor (SHARP-5030i, Thermo Fisher Scientific
Inc, USA), while $PM_{10}$ mass data were obtained from the China National Environmental Monitoring
Center (https://quotsoft.net/air/). During the field campaign, fifty-seven VOC canister samples were
collected at 2-hour intervals from 9:00–19:00 local time on pollution episode days and at 6-hour
intervals from 9:00–21:00 on non-episode days. These VOC samples were subsequently analyzed
using gas chromatography and mass spectrometry (TT24xr, Markes, UK; GC–MS, Thermo Fisher
Scientific Inc, USA) (Liu et al., 2021). A wide-range particle spectrometer (WPS, Model 1000XP,
MSP, USA) was employed to determine the atmospheric particle number size distributions from 10
nm to 10 μm. Taking into account the hygroscopic growth, the relative humidity-adjusted aerosol
surface area concentration (Sa) was calculated based on the determined particle number size
distributions. 95 offline particulate samples were collected every 3-hour interval from 7:00–19:00 and
12-hour intervals from 19:00–7:00 utilizing a high-volume air-sampling system (TE-5170, Tisch
Environmental Inc, USA). The inorganic compositions of the samples, including $Cl^-$, $NO_3^-$, $SO_4^-$,
$NH_4^+$, $Na^+$, $K^+$, $Mg^{2+}$, and $Ca^{2+}$, were determined via ion chromatography (Dionex ICS-600, Thermo
Fisher Scientific Inc, USA). Meteorological data, including temperature, RH, pressure, wind speed,
and wind direction, were monitored by an ultrasonic integrated weather station (RS-FSXCS-N01-1).




This study distinguishes between the "sea case" and the "land case" by analyzing the backward
trajectory of the air mass. Specifically, considering the short lifetime of HONO, the MeteoInfo model
(Wang, 2012) was used to calculate 6-hour air mass backward trajectories starting at the height of 200
meters above ground level, using meteorological parameters from the Global Data Assimilation
System (GDAS, ftp://arlftp.arlhq.noaa.gov/). The criteria for differentiating between the "sea case"
and "land case" is based on the time spent over land or sea during the 6-hour backward air mass, with
cases that spent less than 1 hour over land designated as a "sea case" (Yang et al., 2021a). Following
this criterion, we selected a total of 18 sea cases and 13 land cases (Table S1). The observation data
for the "sea case" and the "land case" were averaged for subsequent analysis.
**2.2 Model setup**
An observation-based chemical box model (OBM) was employed to explore the HONO budgets
and atmospheric oxidizing capacity. The chemical mechanism used for the modeling was obtained
from the Master Chemical Mechanism (MCM) website (http://mcm.york.ac.uk/) and was based on the
MCM v3.3.1 as proposed by Jenkin et al. (2015). The model was constrained with data including
HONO, $O_3$, NO, $NO_2$, $SO_2$, CO, VOCs, $pNO_3^-$, Sa, temperature, RH, pressure, and $JNO_2$. These
variables were averaged and interpolated to a time resolution of 5 minutes, except for VOCs and $pNO_3^-$
which were recorded at 1-hour intervals. The calculation of the photolysis rate of $NO_2$, $JNO_2$, was
determined using Equation 1:

$$JNO_2 = JNO_{2(TUV)} \times \frac{UV_{observed}}{UV_{TUV}} \tag{E1}$$

where $JNO_{2(TUV)}$ and $UV_{TUV}$ are obtained from the Tropospheric Ultraviolet and Visible (TUV)
radiation model (http://cprm.acom.ucar.edu/Models/TUV/Interactive_TUV/). The $UV_{observed}$ was
obtained from the NASA GES DISC (https://disc.gsfc.nasa.gov/). Other photolysis frequencies were
calculated in the OBM and scaled by $JNO_2$. The time series of $JNO_2$ is presented in Figure S1. The
model was pre-run for 1 day to stabilize the simulation of unconstrained species.
In the MCM v3.3.1, the formation of HONO is originally attributed to a homogeneous reaction,
specifically NO + OH → HONO. This study extends the existing mechanism by incorporating
additional sources of HONO into the chemical model. A description of these sources and their
associated mechanisms is provided in the following, and the corresponding parameters are listed in
Table 1.
**Description of HONO sources and sinks adopted in the OBM**
**Direct emission**



In the atmosphere, HONO can be directly released through the exhaust emissions of various
sources. The HONO/NO$_x$ emission ratio, which typically averages around 0.8%, is a common
parameter used to gauge the impact of these vehicular emissions on HONO concentration (Kleffmann
et al., 2003). However, the ratio can fluctuate between 0.3% and 1.6%, depending on engine and fuel
types (Kurtenbach et al., 2001). Prior research indicates that direct emissions contribute significantly
to HONO concentration in urban settings (Zhang et al., 2019; Kramer et al., 2020). However, in rural
and background areas, the vehicular contribution is comparatively insignificant (Liu et al., 2019b; Xue
et al., 2022). Consequently, the contribution of vehicle emissions to HONO is not constant and varies
based on the environment and traffic density.
**Homogeneous reaction of OH + NO → HONO**

$$NO + OH + M \rightarrow HONO + M \tag{R1}$$

The reaction of NO + OH is considered an important gas-phase reaction for HONO formation,
particularly during pollution periods when concentrations of NO and OH are high (Gu et al., 2022a).
We employed the box model to calculate the reaction rate using complex rate coefficients from the
MCM website (http://mcm.york.ac.uk/parameters/complex.htt).
**Heterogeneous reaction of NO$_2$ on aerosol surfaces**

$$NO_2 + H_2O \xrightarrow{\text{aerosol surface}} HONO + HNO_3 \tag{R2}$$

$$NO_2 + hv \xrightarrow{\text{aerosol surface}} HONO \tag{R3}$$

$$k_{aerosol} = 0.25 \times v_{NO_2} \times Sa \times \gamma_a \qquad \qquad \gamma_a = 8 \times 10^{-6} \tag{E2}$$

$$k_{aerosol,\, hv} = 0.25 \times v_{NO_2} \times Sa \times \gamma_{a,\, hv} \times \frac{JNO_2}{JNO_{2,noon}} \qquad \gamma_{a,\, hv} = 4 \times 10^{-5} \tag{E3}$$

$$v_{NO_2} = \sqrt{\frac{8RT}{\pi M}} \tag{E4}$$

The heterogeneous conversion of NO$_2$ on surfaces is a significant source of HONO in the atmosphere.
As illustrated by equations R2 and R3, NO$_2$ reacts with water and light on aerosol surfaces to produce
HONO. The HONO formation rate from heterogeneous reactions is typically first-order with respect
to NO$_2$ concentration (Aumont et al., 2003), and the reactivity of NO$_2$ is known to be significantly
enhanced under irradiated conditions compared to darkness (Yu et al., 2022a). In this study, the uptake
coefficients of NO$_2$ on the aerosol surface in dark and irradiated conditions, $\gamma_a$ and $\gamma_{a,\, hv}$, were set to
$8 \times 10^{-6}$ and $4 \times 10^{-5}$ (Lelièvre et al., 2004; Vandenboer et al., 2013b), respectively. The molecular speed
of NO$_2$ ($v_{NO_2}$, m s$^{-1}$) was calculated using Equation 4, where R represents the ideal gas constant, 8.314
J mol$^{-1}$ k$^{-1}$, T is the absolute temperature (K), and M is the relative molecular weight of NO$_2$ (g mol$^-$



[1]). Sa is the surface area concentration ($m^2\ m^{-3}$) estimated from particle number concentrations
measured by the WPS.

**Heterogeneous reaction of $NO_2$ on ground surfaces**

$$NO_2 + H_2O \xrightarrow{\text{ground surface}} HONO + HNO_3 \tag{R4}$$

$$NO_2 + h\nu \xrightarrow{\text{ground surface}} HONO \tag{R5}$$

$$k_{\text{ground}} = 0.25 \times v_{NO_2} \times \gamma_{g,\,h\nu} \times \frac{S}{V} \qquad\qquad \gamma_g = 1 \times 10^{-6} \tag{E5}$$

$$k_{\text{ground},\,h\nu} = 0.25 \times v_{NO_2} \times \gamma_{g,\,h\nu} \times \frac{S}{V} \times \frac{JNO_2}{JNO_{2,\text{noon}}} \qquad \gamma_{g,\,h\nu} = 2 \times 10^{-5} \tag{E6}$$

$$\frac{S}{V} = \frac{1.7}{BLH} \tag{E7}$$

Equations 5 and 6 delineate the parameterizations for the heterogeneous reaction of $NO_2$ on the ground
surfaces, both in the absence and presence of light. The uptake coefficients of $NO_2$ on the ground
surface under dark and irradiated conditions, $\gamma_g$ and $\gamma_{g,\,h\nu}$, respectively, were set to $1 \times 10^{-6}$ and $2 \times 10^{-5}$
(Kleffmann et al., 1998; Stemmler et al., 2006), respectively. Under ambient conditions, the relative
importance of gas uptake on ground and aerosol surfaces is uncertain, with the influence of land use
categories and chemical compositions (Li et al., 2019). The surface-to-volume ratio, $\frac{S}{V}$, is calculated
by an effective surface of $1.7\ m^2$ per geometric surface in Equation 7 (Vogel et al., 2003). Within the
model, the boundary layer height, BLH, is projected to increase from 300 m at dawn to 1500 m at
14:00 and then decrease back to 300 m at dusk (Xue et al., 2014).

**Photolysis of particulate nitrate**

$$pNO_3^- + h\nu \rightarrow HONO \tag{R6}$$

$$k = \frac{J(pNO_3^-)}{JHNO_{3,\text{noon}}} \times JHNO_{3(MCM)} \tag{E8}$$

In Equation 8, the photolysis rate constant of gaseous $HNO_3$ at noon, $JHNO_{3,\text{noon}}$, is chosen to be ~
$7 \times 10^{-7}\ s^{-1}$ based on previous studies (Ye et al., 2016b). $JHNO_{3(MCM)}$ is calculated by the box model.
Recent research has shown that the photolysis rate of particulate nitrate is significantly faster than that
in the gas and aqueous phases (Zhou et al., 2003; Ye et al., 2016a). We adopt a median value of $8.3 \times 10^{-5}\ s^{-1}$
in our simulation based on a range provided by Ye et al. (2017). Considering the uncertainty of
the parameter values of the above-mentioned HONO formation mechanisms, we conducted the
sensitivity tests with lower and upper values in Sections 3.2 and 3.3.

**Photolysis of HONO**

$$HONO + h\nu \rightarrow OH + NO \tag{R7}$$





The primary loss pathway of HONO is through photolysis following sunrise, which significantly
contributes to the atmospheric OH budget. The photolysis rate of HONO, J(HONO), in the OBM, is
constrained by $JNO_2$.
**Homogeneous reaction between HONO and OH**

$$HONO + OH \rightarrow H_2O + NO_2 \tag{R8}$$

The relevant kinetic parameter of the reaction between HONO and OH is available from the MCM
mechanism, and its reaction rate coefficient is dependent solely on the temperature.
**Dry deposition of HONO**

$$k = \frac{v_{HONO}}{BLH} \tag{E9}$$

Here, $v_{HONO}$ is the dry deposition velocity of HONO (cm s$^{-1}$). Harrison and Kitto (1994) suggested the
range of $v_{HONO}$ was 0.2–1.7 cm s$^{-1}$, and a value of 1.0 cm s$^{-1}$ was employed in this study.
**3 Results and discussion**
**3.1 Concentration levels and temporal variations**

Figure 2 displays the time series of HONO, HONO/$NO_2$, $NO_x$, $O_3$, CO, $SO_2$, $PM_{2.5}$, and $pNO_3^-$,

along with meteorological parameters (i.e., temperature, RH, and wind) measured throughout the field
campaign. The presence of missing data in the time series resulted from instrument maintenance and
calibration. Instrument maintenance and calibration resulted in gaps in the time series data. The
observation site underwent dust periods on April 27–28 and May 7–8, as well as periods of
photochemical pollution on May 5–6, 13, and 17–18. In this study, a photochemical pollution period
is classified as a day when the maximum daily 8-hour average $O_3$ concentration (MDA8$O_3$) exceeds
75 ppbv(the Grade II National Ambient Air Quality Standard). A dust period is recognized when the
peak $PM_{10}$ concentration surpasses 150 µg m$^{-3}$, and the $PM_{2.5}/PM_{10}$ ratio falls below 0.4, based on
previous research (Liu et al., 2006; Wu et al., 2020). Section 3.2 provides a comprehensive explanation
of the differences in pollutant concentrations and HONO budgets during dust and photochemical
pollution periods.

Table 2 summarizes the descriptive statistics of the species and meteorological parameters

measured during the observation period. The average (± standard deviation, SD) temperature and RH
were 15.1 ± 3.4 °C and 68.7 ± 26.1%, respectively, indicating a moderate spring temperature and
relatively high RH influenced by marine air masses. The primary pollutant concentrations were
relatively low, as indicated by the mean mixing ratios of 0.9 ± 1.7 ppbv, 5.9 ± 4.8 ppbv, 284.0 ± 118.8
ppbv, and 1.0 ± 0.8 ppbv for NO, $NO_2$, CO, and $SO_2$, respectively. These low levels suggest Mount



Lao is a relatively clean site with minimal impact from nearby anthropogenic sources. The high $O_3$
concentration (60.4 ± 15.8 ppbv) implies that photochemical reactions were relatively strong during
observation. The mean concentrations of $PM_{2.5}$ and $pNO_3^-$ were 21.2 ± 21.09 µg m$^{-3}$ and 4.6 ± 5.0 µg
m$^{-3}$, respectively.

During the campaign, the mean concentration of HONO was 0.46 ± 0.37 ppbv, with a maximum

mixing ratio of 3.14 ppbv recorded at 17:00 on May 4th. The concentration level of HONO at Mount
Lao is lower than at urban sites with higher $NO_2$ concentrations (Li et al., 2018; Yu et al., 2022b; Hao
et al., 2020). However, it is notably higher than other clean coastal and remote marine sites, as Table
S2 illustrates (Zhu et al., 2022; Zha et al., 2014; Meusel et al., 2016; Crilley et al., 2021; Villena et al.,
2011). Past research conducted in urban and rural areas found that the HONO/$NO_2$ ratio, which
indicates the extent of $NO_2$ conversion to HONO, typically ranges from 0.02 to 0.08 (Jiang et al., 2022).
The higher HONO/$NO_2$ value (0.13) measured at Mount Lao highlights the potentially significant role
of non-$NO_x$ related HONO sources or higher heterogeneous conversion of $NO_2$ efficiency at this site.

Figure 3 illustrates the average diurnal patterns of HONO and related species. The diurnal cycle

of CO and $SO_2$ is similar, with peak concentrations observed during midday and relatively stable
concentrations during nighttime. The concentration of $O_3$ increases with the accumulation of
photochemical generation during the afternoon and decreases steadily after sunset. Contrary to most
urban or rural locations, the concentration of HONO at Mount Lao peaks at noon, similar to remote
areas (Jiang et al., 2022; Ye et al., 2016a). NOx, comprising NO and $NO_2$, shows a similar temporal
variation trend to HONO, suggesting potential photolytic sources for them (Reed et al., 2017). During
the daytime (7:00–17:00), the average concentration of HONO was 1.56 times higher than at night
(17:00–7:00), with concentrations of 0.54 ppbv during the day and 0.35 ppbv at night. Given the short
lifetime of HONO during the day—only a matter of minutes—the noon HONO peak concentration
suggests an in situ photochemical source for HONO (Kasibhatla et al., 2018). The ratio of HONO to
$NO_2$ shows an increasing trend until sunrise, suggesting heterogeneous conversion from $NO_2$ to HONO
during nighttime. However, unlike urban areas where the ratio of HONO to $NO_2$ decreases during the
daytime (Zhang et al., 2019; Gu et al., 2022a), the ratio even increases during the midday period at
Mount Lao, implying that HONO from sources other than $NO_2$ conversion also significantly
contributes to HONO concentration (Yang et al., 2021a). Considering that the influence of HONO on
the OH radical and $O_3$ is primarily observed during the daytime, the higher concentration of HONO
during the daytime at the Mount Lao site suggests the presence of strong daytime HONO sources. The
primary objective of our following study is to analyze the daytime budgets of HONO in the coastal
atmosphere of Mount Lao.



**3.2 Daytime HONO budgets in dust and photochemical pollution periods**

The daytime HONO budgets were examined during periods of dust and photochemical pollution using an updated OBM, with the aim of assessing whether our current understanding of HONO sources is sufficient to explain observed concentrations. Table 4 presents the mean daytime concentrations of HONO and other species during the dust, photochemical pollution, and non-polluted periods (i.e., days devoid of dust and photochemical pollution). On average, the daytime HONO concentrations during dust and photochemical pollution periods were $0.57 \pm 0.39$ ppbv and $0.44 \pm 0.29$ ppbv, respectively. The dust period exhibited significantly higher concentrations of $NO_2$, $PM_{2.5}$, $pNO_3^-$, and Sa, with increased factors of 1.4, 2.6, 2.3, and 2.3, respectively, compared to the non-polluted period. During the photochemical pollution period, the daytime mean values of $O_3$, CO, $SO_2$, $PM_{2.5}$, $pNO_3^-$, and $JNO_2$ were 78.8 ppbv, 353.8 ppbv, 1.7 ppbv, 25.0 μg m$^{-3}$, 6.2 μg m$^{-3}$, and $7.0 \times 10^{-3}$ s$^{-1}$, respectively. These values were approximately 1.4, 1.3, 2.4, 1.5, 2.1, and 1.6 times higher than those during the non-polluted period.

Figure 4 compares the observed and modeled HONO concentrations during dust and photochemical pollution periods and illustrates the contribution of various sources and sinks to the HONO budget. The study examined two scenarios: the base case, which only considered the homogeneous reaction NO + OH, and the model case, which considered all seven HONO sources outlined in Table 1. The results indicated that the base case significantly underestimated the HONO concentration, consistent with previous studies (Liu et al., 2019b; Zhu et al., 2022). However, the model case effectively replicated the observed HONO concentrations for both periods, even the high noon concentrations. The index of agreement (IOA) values for HONO during the dust and photochemical pollution periods were 0.96 and 0.88, respectively. This suggests that the updated parameterization scheme employed in the model can adequately account for the observed HONO concentrations at Mount Lao.

During the daytime, the average modeled production rates of HONO were 1.66 ppbv h$^{-1}$ and 0.90 ppbv h$^{-1}$ for the dust and photochemical pollution periods, respectively. The maximum HONO production rate was significantly higher during the dust period (3.50 ppbv h$^{-1}$) compared to the photochemical pollution period (1.69 ppbv h$^{-1}$) and was even comparable to levels observed during haze periods at polluted urban or rural sites (Xue et al., 2020; Gu et al., 2022a; Zhang et al., 2019).

Based on the model results of detailed HONO budgets, the dominant pathway for daytime HONO production during the dust period was photo-enhanced heterogeneous conversion of $NO_2$ on the aerosol surface, accounting for 53% ($0.87 \pm 0.66$ ppbv h$^{-1}$) of the simulated daytime HONO production



rate. Wang (2003) reported sudden increases in HONO concentration during nocturnal dust storm
events and observed a higher ratio of HONO to $NO_2$ (0.18). The enhanced efficiency of $NO_2$ to HONO
on mineral dust particles suggests a potentially significant impact of dust aerosol on nitrogen
compound distribution. Further research is needed to understand the contribution of dust to HONO
formation and nitrogen cycling during the daytime, as well as its global impact. For the photochemical
pollution period, the major sources of HONO included the photolysis of particulate nitrate, the photo-
enhanced heterogeneous conversion of $NO_2$ on the aerosol surface, and the homogeneous reaction of
OH and NO, which contributed 34%, 27%, and 27% of the daytime HONO production rate,
respectively. This points to the significant role of photochemical processes under intense solar radiation.
Direct emissions had a negligible contribution during both the dust and photochemical pollution
periods, accounting for less than 2%. The photolysis of HONO was the dominant loss pathway
throughout the day for all measurement periods, accounting for more than 90% of HONO sinks.
The model was subjected to sensitivity tests by increasing or decreasing selected parameters by
factors of 5 and 2 (Table S4, Figures S2, and S3). Even with such a broad range of parameter variation,
the heterogeneous reaction of $NO_2$ on aerosols and the photolysis of nitrate to form atmospheric HONO
remained significant sources of HONO under both dust and photochemical pollution periods. This
suggests that our current understanding of HONO sources, based on existing mechanisms, can
generally explain the observed concentrations of HONO. However, it is important to note that
differences in parameter selection can significantly affect the relative contributions of each pathway.
Given the considerable uncertainties in the uptake coefficient of $NO_2$ and the enhancement factors of
photolysis of nitrate, further experimental studies are necessary to evaluate their effects on HONO in
different environmental conditions.

**3.3 Missing daytime HONO source in "sea case"**

Recent field studies suggest potential unidentified daytime sources of nitrous acid (HONO) in the
marine atmosphere, with high daytime HONO levels recorded (Yang et al., 2021a; Ye et al., 2016a).
Figure 5 shows the diurnal variation of the selected "sea case" and "land case", with corresponding
statistical results in Table S3. In the "sea case", daytime concentrations of typical primary pollutants,
such as CO and $SO_2$, are significantly lower than those in the "land case" (251 ± 59 ppbv vs. 335 ±
115 ppbv and 0.7 ± 0.4 ppbv vs. 1.4 ± 0.8 ppbv for CO and $SO_2$, respectively). Concurrently, the "sea
case" shows a lower daytime temperature (15.2 ± 3.0°C vs. 18.7 ± 3.8°C) and higher RH (76.3 ± 25.9%
vs. 47.3 ± 20.3%) compared to the "land case". This is consistent with our understanding of marine air
masses, which tend to be cleaner and more humid. These findings validate our classification method
of "land case" and "sea case" based on the backward air mass trajectory.





332   Secondary pollutants in marine air masses, such as $O_3$, $PM_{2.5}$, and $pNO_3^-$, also register lower

333 daytime concentrations than in the "land case" (59.4 ± 10.3 ppbv vs. 63.4 ± 13.3 ppbv, 13.2 ± 5.8 µg

334 $m^{-3}$ vs. 29.9 ± 22.8 µg $m^{-3}$, and 1.3 ± 0.5 µg $m^{-3}$ vs. 10.0 ± 3.3 µg $m^{-3}$, respectively). Though the

335 HONO concentration in marine air masses is less than that in the "land case" (0.42 ± 0.25 ppbv vs.

336 0.51 ± 0.22 ppbv), it maintains a relatively high level, particularly during intense photolysis periods

337 around noon when the HONO concentration in the "sea case" marginally increases. $NO_x$

338 concentrations in the "sea case" are also lower than in the "land case", but the difference is less

339 substantial than primary pollutants, with both NO and $NO_2$ showing concentration peaks around noon.

340 Nighttime observations in the "sea case" show a higher $HONO/NO_2$ ratio (0.12), which has been noted

341 in earlier studies (Zha et al., 2014), suggesting strong nocturnal HONO formation in marine air masses.

342 Here, we focus on the sources of HONO during the day under the influence of marine air masses.

343 Utilizing the updated chemical model, we examine the HONO budgets in both "sea" and "land" cases.

344 In the "land case", the simulated HONO concentration aligns well with the observed HONO

345 concentration, with a high index of agreement (IOA) value of 0.94 (Figure 6a). The peak HONO

346 production rate observed at Mount Lao (2.69 ppbv $h^{-1}$) surpasses that calculated in continental air

347 masses at Hok Tsui, Hong Kong (less than 1.5 ppbv $h^{-1}$) (Gu et al., 2022b). The contributions of photo-

348 enhanced heterogeneous reactions of $NO_2$ on the aerosol surface (22%) and photolysis of $pNO_3^-$ (20%)

349 are comparable (Figure 6c). Model results reveal that the homogeneous reaction between NO and OH

350 is the predominant HONO formation pathway, contributing an average of 44% (0.52 ± 0.38 ppbv $h^{-1}$).

351 Despite a lower absolute rate than in urban areas, the relative contribution is significant(Gu et al.,

352 2022a; Yu et al., 2022b). This result suggests that similar to the findings in the cases of dust and

353 photochemical pollution, the current model's parameterization reasonably accounts for the observed

354 HONO concentration in the "land case".

355   However, in the "sea case", while the updated model has improved in simulating HONO

356 concentrations, with an average concentration increase from 0.05 ppbv to 0.11 ppbv, it falls short of

357 the observed concentration (0.42 ppbv), indicating a substantial unidentified HONO source. At noon,

358 the missing HONO production rate ($P_{missing}$) can reach up to 0.70 ppbv $h^{-1}$. This value is slightly higher

359 than the result calculated by Meusel et al. (2016) on Cyprus Island (about 0.5 ppbv $h^{-1}$), but lower than

360 that reported by Yang et al. (2021a) in coastal Qingdao (up to 1.83 ppbv $h^{-1}$, including all non-NO+OH

361 pathways). Sensitivity tests were conducted to assess the impact of parameter selection on simulation

362 results, but even with much larger parameters (Table S4), the model fails to explain the observed

363 HONO concentrations (Figure S4).



The correlation analysis reveals that the missing HONO production rate correlates strongly with
$JNO_2$ and $JNO_2 \times pNO_3^-$ (Figure S6), with correlation coefficients (r) of 0.90 and 0.73, respectively.
This indicates that the missing HONO sources are closely related to photochemical processes. This
concurs with recent multi-site HONO analysis results, which propose a significant role of
photochemical processes in observed HONO concentrations in remote areas (Jiang et al., 2022). We
postulate that all missing HONO originates from photochemical processes and have calculated the
required enhancement factors (EF) for nitrate photolysis rates (Text S2) and the uptake coefficient
required for $NO_2$ on aerosol and ground surfaces (Figure S5). To account for the observed HONO
concentrations, the required EF is approximately 4000. While Andersen et al. (2023) noted that the EF
increases with decreasing nitrate concentration, a 4000-fold difference exceeds all laboratory and field
observations to date (Ye et al., 2017; Andersen et al., 2023). The required uptake coefficient of $NO_2$
on aerosol and ground surface reach $4 \times 10^{-4}$, exceeding previous laboratory studies (Liu et al., 2019a;
Stemmler et al., 2007). This suggests that the observed missing HONO source in the "sea case" cannot
be explained by the current photochemical processes. This deviates from the findings of Zhu et al.
(2022), who discovered that nitrate photolysis could explain the observed HONO concentrations in
clean marine air masses using a moderate EF of 29. In recent years, many observations have noted
distinct HONO characteristics under the influence of marine air masses, differing from those in
continental air masses, but the specific mechanisms are still lacking (Yang et al., 2021a; Zhu et al.,
2022; Crilley et al., 2021; Meusel et al., 2016). The ocean surface contains abundant nitrogen-
containing substances (e.g., dissolved nitrate, ammonia, aliphatic amine, dissolved free amino acids)
(Donaldson and George, 2012; Altieri et al., 2016), encompassing both organic and inorganic nitrogen.
This is particularly true in polluted coastal areas where surface nitrogen content is rich. It merits
investigation whether these nitrogen-containing substances in the alkaline sea-surface microlayer can
directly affect HONO production or enhance HONO formation by photolysis on the formed sea salt
aerosols. Additionally, the presence of halogens in oceanic air masses might promote nitrate photolysis
(Zhang et al., 2020).

## 390  3.4 Impacts of HONO on $O_3$ and OH production

To quantify the impact of HONO, especially in the marine atmosphere, on $O_3$ and OH radicals,
we conducted further scenario simulations using a chemical box model. In the "with HONO" scenario,
we input the observed HONO concentrations to constrain the model. In contrast, in the "without
HONO" scenario, the model set HONO concentrations to zero. The differences between these
scenarios illustrate the impact of HONO chemistry on $O_3$ and OH radicals in the atmosphere. To further
investigate the effect of the missing HONO sources in marine air masses, we established a third



simulation scenario, "without missing HONO", in the marine air mass simulation. In this scenario, the
model includes the latest HONO formation mechanisms discussed earlier but without the constraint of
observed HONO.

HONO significantly influences the production of $O_3$ and OH radicals, regardless of whether the

overall situation during the observation period ("overall case") or the situation within the marine air
masses ("sea case") is considered (Figure 7). The net $O_3$ production rate can be determined by the
difference between the $O_3$ production rate ($P(O_3)$) and loss rate ($L(O_3)$) (Xue et al., 2014). Specifically,
the absence of HONO resulted in a decrease in the $O_3$ and OH radical production rates in the "overall
case" from 7.39 ppbv $h^{-1}$ and $1.44 \times 10^7$ molecules $cm^{-3}$ $s^{-1}$ to 2.99 ppbv $h^{-1}$ (a 59% reduction) and
$2.78 \times 10^6$ molecules $cm^{-3}$ $s^{-1}$ (an 81% reduction), respectively. Similarly, in the marine air masses, the
$O_3$ and OH radical production rates decreased from 6.22 ppbv $h^{-1}$ and $7.69 \times 10^6$ molecules $cm^{-3}$ $s^{-1}$ to
3.20 ppbv $h^{-1}$ (a 49% reduction) and $2.11 \times 10^6$ molecules $cm^{-3}$ $s^{-1}$ (a 73% reduction), respectively.
Regarding concentration, the absence of HONO chemistry resulted in a reduction in the average $O_3$
concentration in the overall situation from 40.4 ppbv to 35.0 ppbv and a reduction in the average OH
radical concentration from $3.6 \times 10^6$ molecules $cm^{-3}$ to $1.7 \times 10^6$ molecules $cm^{-3}$. Furthermore, the peak
concentrations of $O_3$ and OH radicals decreased by 15% and 53% (from 59.3 ppbv to 50.3 ppbv and
from $5.2 \times 10^6$ molecules to $2.4 \times 10^6$ molecules $cm^{-3}$), respectively. These findings are consistent with
previous observational studies (Gu et al., 2022a; Yang et al., 2021a), highlighting the significant
impact of HONO on $O_3$ and OH radicals.

The simulated $O_3$ and OH radical concentrations for marine air masses also significantly decrease

when the missing HONO source is not considered ("without missing HONO"). Specifically, the
average $O_3$ concentration decreased from 29.1 ppbv to 26.8 ppbv (an 8% reduction), while the average
OH radical concentration decreased from $2.5 \times 10^6$ molecules $cm^{-3}$ to $1.8 \times 10^6$ molecules $cm^{-3}$ (a 30%
reduction). The peak concentrations of $O_3$ and OH radicals decreased by 8% and 28% (from 40.8 ppbv
to 37.4 ppbv and from $3.4 \times 10^6$ molecules $cm^{-3}$ to $2.4 \times 10^6$ molecules $cm^{-3}$), respectively. Regarding
the relative contribution to OH radicals, HONO accounts for 79% and 55% in the "overall case" and
"sea case", respectively, both exceeding the combined contribution of other pathways (photolysis of
$O_3$ contributes 14% and 25%, respectively). Notably, in the "sea case", if the observed values are not
input as constraints, and only the updated mechanisms are used, the model still significantly
underestimates the impact of HONO on $O_3$ and OH radicals. Given the relatively limited observational
data on HONO in coastal or marine areas and the unclear understanding of the missing HONO sources
in the ocean, the impact of marine emissions on atmospheric oxidizing capacity may be significantly



underestimated. This underscores the importance of further research in this area to enhance our
understanding of the role of HONO in atmospheric chemistry, especially in marine environments.
**4 Conclusions**
This study presents a comprehensive investigation of the characteristics and sources of nitrous
acid (HONO) in the coastal environment of Qingdao. The analysis utilizes observational data from the
Mount Lao site in Qingdao during spring 2021 and an updated chemical box model that integrates
HONO mechanisms. The focus lies on discerning the unidentified HONO sources in marine air masses
and comprehending their effects on atmospheric chemistry, emphasizing $O_3$ and OH concentrations.
Despite a relatively pristine coastal atmosphere, HONO concentrations are considerably higher
than previously thought ($0.46 \pm 0.37$ ppbv), notably during daytime. This observation persists in lower
primary pollutant concentrations such as CO and $SO_2$ within marine air masses, suggesting missing
HONO sources tied to photochemical processes. An updated chemical model's simulation reveals that
the mechanisms behind HONO formation can satisfactorily account for the observed HONO
concentrations during the dust and photochemical pollution periods. Yet, in marine scenarios, the
model falls short of matching observed concentrations, pointing to a strong unidentified HONO source
within the marine atmosphere. Sensitivity tests and correlation analyses emphasize the importance of
photochemical processes in unidentified HONO sources. Nevertheless, if these unknown sources are
attributed to either nitrate photolysis or heterogeneous $NO_2$ reactions, the necessary nitrate photolysis
rate and the heterogeneous uptake coefficient of $NO_2$ would surpass the upper thresholds established
by current laboratory studies. In light of these findings, future research must target uncovering the
mechanisms behind the missing HONO sources in marine air masses. Specifically, the role of nitrogen-
containing substances at the ocean's surface and the potential influence of halogens in promoting
nitrate photolysis warrant further examination.



**Acknowledgments**

This work was supported by the National Natural Science Foundation of China (grants nos. 42061160478 and 42105106) and the National Key Research and Development Programme of the Ministry of Science of Technology of China (grant no. 2022YFC3701101). We would like to express our gratitude to the University of Leeds for providing the Master Chemical Mechanism and to NCAR for the Tropospheric Ultraviolet Visible (TUV) radiation model. We also thank Yaqiang Wang for developing the open-source software MeteoInfo.

**Author contributing**

LX and HS conceptualized the research. XZ drafted the initial manuscript and analyzed the data. CD supported funding the observation. YZ, CD, and XW designed the field campaign. MZ, JZ, YS, YL, YS, HL, and JM conducted the field campaign. JZ and YL analyzed the aerosol samples and VOC samples, respectively. MZ and YZ assisted with the model simulation. YY, YN, and JT contributed to figure creation. LX, HS, MG, and WW revised the original manuscript.

**Competing interests**

The authors declare that they have no conflict of interest.

**Data availability**

The data supporting this study are available upon request from the corresponding author.



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



**Figures and Tables**

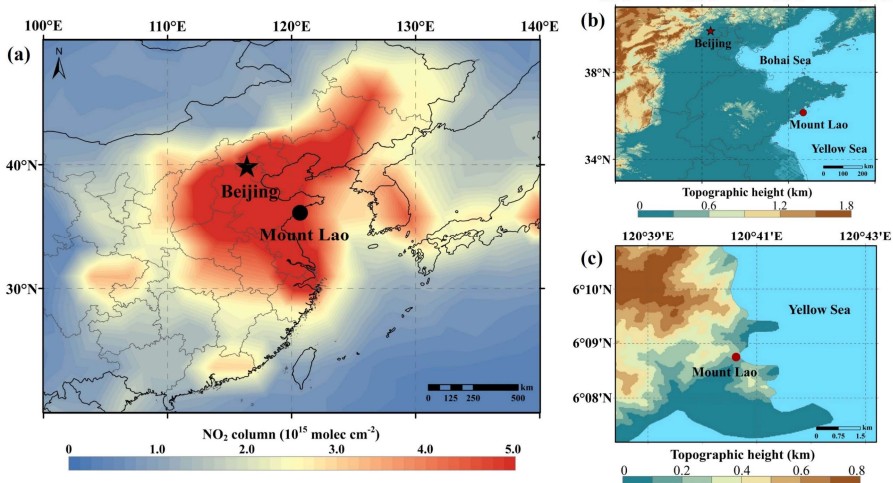

**Figure 1.** Maps showing the location of the monitoring site. Figure 1a is colored by tropospheric $NO_2$ column density in May 2021 from the Ozone Monitoring Instrument (OMI, https://www.earthdata.nasa.gov/), and Figure 1b and Figure 1c are colored by the geographical height from the Geospatial Data Cloud (http://www.gscloud.cn/).





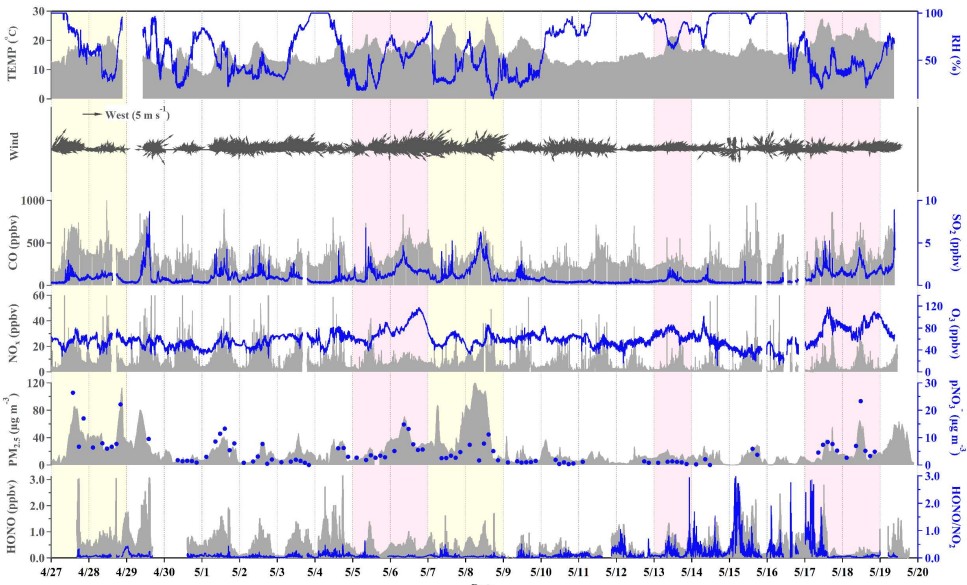

**Figure 2.** Time series of HONO, meteorological parameters, and related species measured during the campaign. The yellow shaded areas correspond to the period of dust, while the pink shaded areas represent the period of photochemical pollution.



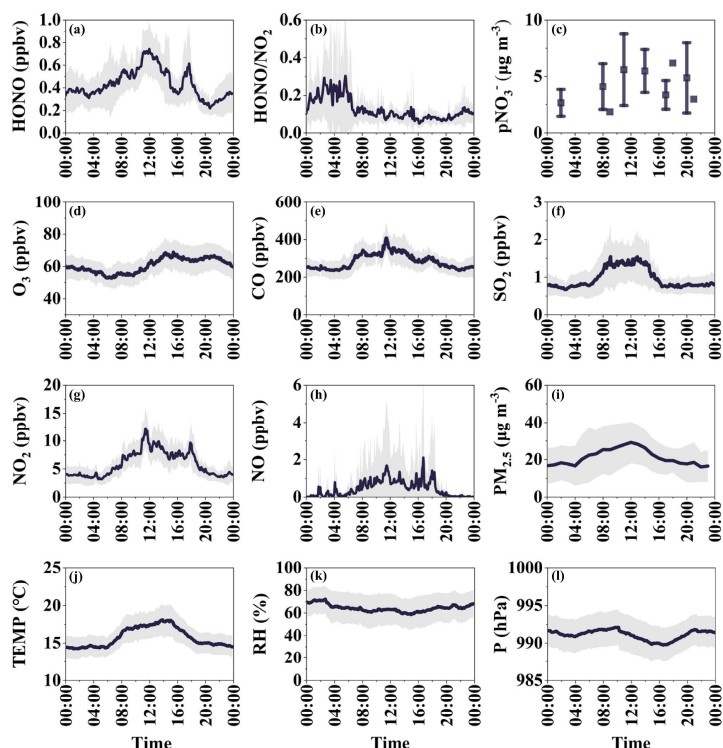

**Figure 3.** Average diurnal variations of (a) HONO, (b) HONO/$NO_2$, (c) particle nitrate, (d) $O_3$, (e) CO, (f) $SO_2$, (g) $NO_2$, (h) NO, (i) $PM_{2.5}$, (j) temperature, (k) RH, and (l) pressure during the observation period. The shaded area indicates the range of half of the standard deviation.



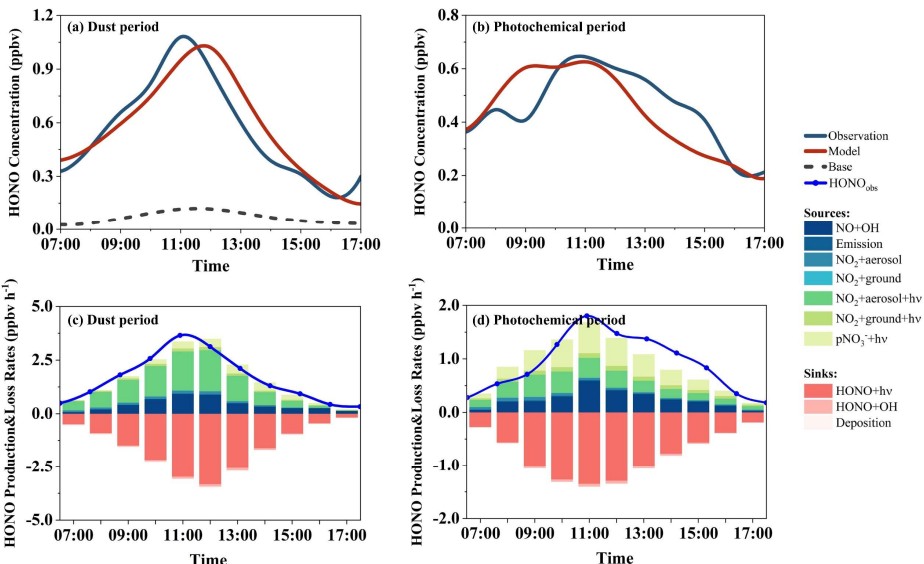


**Figure 4.** Daytime HONO budgets in dust (a, c) and (b, d) photochemical period at Mount Lao. The
base case only considered the homogeneous reaction of NO + OH, and the model case considered the
updated HONO sources described in this study.





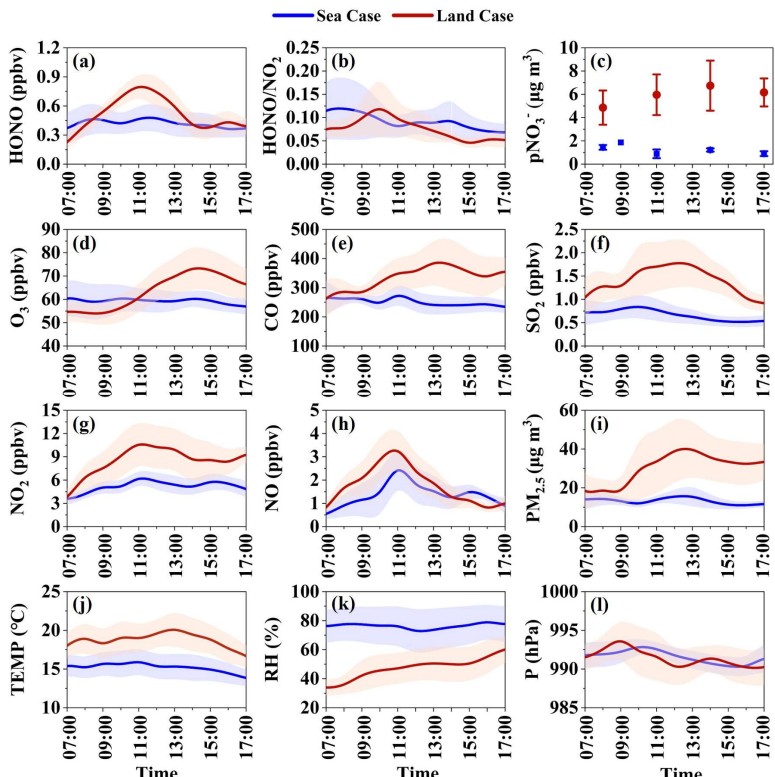


**Figure 5.** Average diurnal variations of HONO and related parameters in the "sea case" and the "land

case" during the campaign at Mount Lao. The shaded area indicates half of the standard deviation.





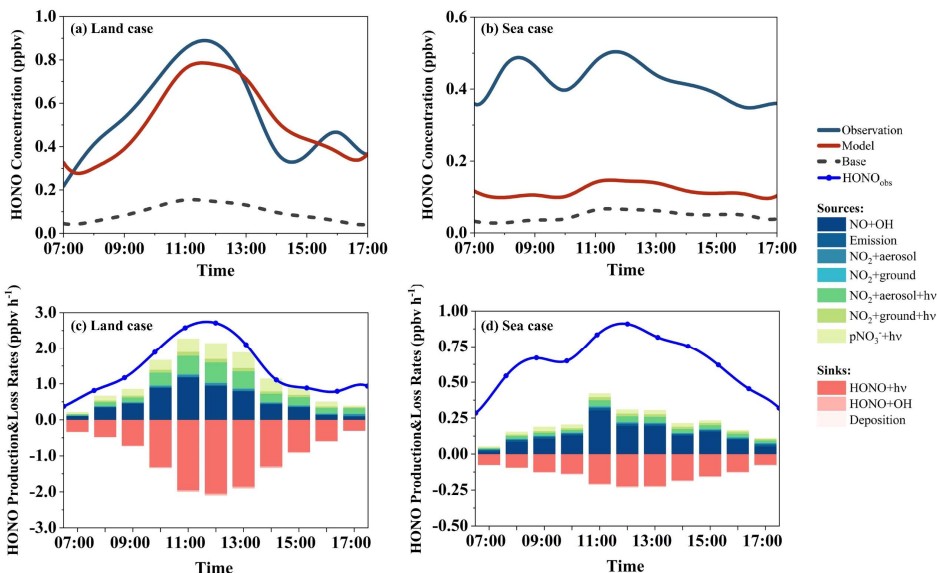

**Figure 6.** Comparison of the observed and modeled daytime (7:00–17:00) HONO concentrations and modeled HONO budgets in the "land case" (a, c) and the "sea case" (b, d).



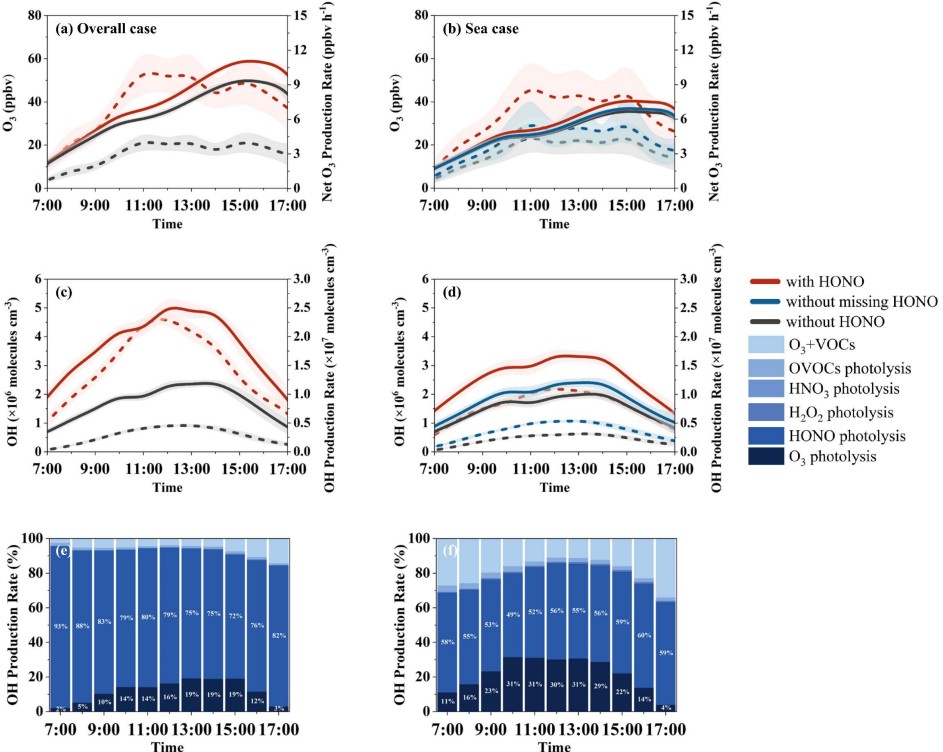

**Figure 7.** Comparison of simulated $O_3$ and OH radical concentration (solid lines) and production rate (dashed lines) with and without HONO measurement data constraints and relative diurnal contributions of different OH radical sources with HONO constrained in the "land case" (a, c, e) and the "sea case" (b, d, f). The shaded area indicates the standard deviation.



**Table 1**. Summary of HONO source and sinks included in the box model.

| Pathways | Parametrization | References |
|---|---|---|
| Direct emission | $k_{emission} = 0.8\%$ | Kleffmann et al. (2003) |
| $OH + NO \rightarrow HONO$ | $k_{OH+NO}$ | Calculated in model |
| $NO_2 + H_2O \xrightarrow{\text{aerosol surface}} HONO + HNO_3$ | $k = 0.25 \times v_{NO_2} \times Sa \times \gamma_a$ $\gamma_a = 8 \times 10^{-6}$ | Vandenboer et al. (2013a) |
| $NO_2 + H_2O \xrightarrow{\text{ground surface}} HONO + HNO_3$ | $k = 0.25 \times v_{NO_2} \times \gamma_g \times \frac{S}{V}$ $\gamma_g = 1 \times 10^{-6}, \frac{S}{V} = \frac{1.7}{BLH}$ | Kleffmann et al. (1998); Vogel et al. (2003) |
| $NO_2 + hv \xrightarrow{\text{aerosol surface}} HONO$ | $k = 0.25 \times v_{NO_2} \times Sa \times \gamma_{a,\,hv} \times \frac{JNO_2}{JNO_{2,noon}}$ $\gamma_{a,\,hv} = 4 \times 10^{-5}$ | Lelièvre et al. (2004) |
| $NO_2 + hv \xrightarrow{\text{ground surface}} HONO$ | $k = 0.25 \times v_{NO_2} \times \gamma_{g,\,hv} \times \frac{S}{V} \times \frac{JNO_2}{JNO_{2,noon}}$ $\gamma_{g,\,hv} = 2 \times 10^{-5}, \frac{S}{V} = \frac{1.7}{BLH}$ | Stemmler et al. (2006); Vogel et al. (2003) |
| $pNO_3^- + hv \rightarrow HONO$ | $k = \frac{8.3 \times 10^{-5}}{7 \times 10^{-7}} \times J(HNO_3)_{MCM}$ | Ye et al. (2017) |
| $HONO + hv \rightarrow NO + OH$ | $k = J(HONO)$ | Calculated in model |
| $HONO + OH \rightarrow H_2O + NO_2$ | $k_{OH+HONO}$ | Calculated in model |
| Deposition | $k = \frac{v_{HONO}}{BLH}$ | Calculated in model |




**Table 2.** Statistics of measured species and meteorological parameters during the campaign.

| Parameters | Mean | SD | Minimum | Median | Maximum |
|---|---|---|---|---|---|
| HONO (ppbv) | 0.46 | 0.37 | < DL (0.005) | 0.38 | 3.14 |
| HONO/$NO_2$ | 0.13 | 0.24 | – | 0.07 | 2.97 |
| NO (ppbv) | 0.9 | 1.7 | 0.1 | 0.2 | 38.3 |
| $NO_2$ (ppbv) | 5.9 | 4.8 | 0.4 | 4.6 | 65.1 |
| $O_3$ (ppbv) | 60.4 | 15.8 | 11.6 | 58.8 | 118.1 |
| CO (ppbv) | 284.0 | 118.8 | 104.2 | 250.3 | 1046.7 |
| $SO_2$ (ppbv) | 1.0 | 0.8 | < DL (0.12) | 0.7 | 8.9 |
| $PM_{2.5}$ ($\mu g\ m^{-3}$) | 21.2 | 21.0 | < DL (0.5) | 14.4 | 120.7 |
| Sa ($m^2\ m^{-3}$) | $6.2 \times 10^{-4}$ | $5.8 \times 10^{-4}$ | $2.8 \times 10^{-4}$ | $4.2 \times 10^{-4}$ | $3.1 \times 10^{-3}$ |
| $pNO_3^-$ ($\mu g\ m^{-3}$) | 4.6 | 5.0 | 0.02[*] | 2.8 | 26.4 |
| TEMP (℃) | 15.1 | 3.4 | 7.5 | 15.6 | 27.9 |
| RH (%) | 68.7 | 26.1 | 9.0 | 64.8 | 99.9 |
| P (kPa) | 991.1 | 4.4 | 979.0 | 991.0 | 1003.0 |
| WS ($m\ s^{-1}$) | 1.23 | 0.96 | 0[*] | 1.00 | 9.30 |
| WD (°) | – | – | 0 | 247 | 354 |

DL: detection limit.



**Table 3**. Comparison of the statistics for the measured species and meteorological parameters during
dust, photochemical pollution, and non-polluted periods in the daytime (7:00–17:00).

| Parameters | Dust period | Photochemical period | Non-polluted period |
|---|---|---|---|
| HONO (ppbv) | $0.57 \pm 0.39$ | $0.44 \pm 0.29$ | $0.40 \pm 0.34$ |
| HONO/NO$_2$ | $0.07 \pm 0.04$ | $0.10 \pm 0.13$ | $0.10 \pm 0.12$ |
| NO (ppbv) | $1.8 \pm 1.8$ | $1.2 \pm 1.4$ | $1.8 \pm 2.0$ |
| NO$_2$ (ppbv) | $9.8 \pm 5.0$ | $7.1 \pm 4.4$ | $7.1 \pm 4.6$ |
| O$_3$ (ppbv) | $58.0 \pm 10.8$ | $78.8 \pm 17.3$ | $54.9 \pm 11.7$ |
| CO (ppbv) | $371.8 \pm 151.9$ | $353.8 \pm 117.5$ | $277.6 \pm 98.0$ |
| SO$_2$ (ppbv) | $1.6 \pm 1.3$ | $1.7 \pm 0.8$ | $0.7 \pm 0.6$ |
| PM$_{2.5}$ ($\mu$g m$^{-3}$) | $45.4 \pm 32.3$ | $25.0 \pm 17.4$ | $17.2 \pm 12.6$ |
| PM$_{10}$ ($\mu$g m$^{-3}$) | $235.3 \pm 200.8$ | $68.0 \pm 47.2$ | $32.8 \pm 21.8$ |
| Sa (m$^2$ m$^{-3}$) | $1.28\times10^{-3} \pm 8.41\times10^{-4}$ | $6.81\times10^{-4} \pm 4.73\times10^{-4}$ | $5.58\times10^{-4} \pm 4.22\times10^{-4}$ |
| pNO$_3^-$ ($\mu$g m$^{-3}$) | $7.0 \pm 6.2$ | $6.2 \pm 5.6$ | $3.0 \pm 3.4$ |
| TEMP (℃) | $19.0 \pm 3.4$ | $20.5 \pm 2.7$ | $15.1 \pm 2.4$ |
| RH (%) | $47.8 \pm 24.7$ | $47.4 \pm 17.2$ | $71.6 \pm 27.0$ |
| WS (m s$^{-1}$) | $0.42 \pm 0.35$ | $0.65 \pm 0.33$ | $0.38 \pm 0.25$ |
| JNO$_2$ (s$^{-1}$) | $6.6\times10^{-3} \pm 2.2\times10^{-3}$ | $7.0\times10^{-3} \pm 2.1\times10^{-3}$ | $4.5\times10^{-3} \pm 2.2\times10^{-3}$ |
