# Peer review of "Nitrous Acid Budgets in Coastal Atmosphere: Potential Daytime Marine Sources"

_EGUsphere, 2023_

## Author Comment (AC1)

**Response to Editors and Reviewers**
We gratefully thank the reviewers for their constructive comments and suggestions to improve the manuscript. As detailed below, the reviewers' comments are shown in *black italic*; our response to the comments is in blue. New or modified text is in red.

**Referee 1:**

*HONO is a very important nitrogen-containing reactive species in the atmospheric environment, which has a very great influence on atmospheric oxidability. At present, there are still many unknowns about the formation mechanisms of HONO, and it is very necessary to carry out HONO-related studies.*

**Response:** Thanks for the reviewer's comments and suggestions. We have addressed the specific comments and revised the manuscript accordingly. For clarity, the reviewer's comments are listed below in *black italic*, while our responses and changes in the manuscript are shown in blue and red, respectively.

*Major comments:*

*1.    The title of this manuscript emphasizes "marine source", but the discussion of the effects of marine source on the observed HONO only takes up a very small part of this study, which in fact only shows that the large difference between the simulated and observed HONO under the "sea case" conditions is unexplained, while the contribution of the marine source and the mechanisms are not studied.*

**Response:** Thanks for your comment. In this study, we found that although the updated chemical box model performed well in simulating daytime HONO production from non-marine sources (photochemical and dust episodes), however, it failed to reproduce HONO production in "sea case" (up to 0.7 ppbv $h^{-1}$). Based on correlation analysis, we suggested that the missing marine sources are strongly associated with solar radiation (Figure S7) and assessed the impact of marine sources on atmospheric oxidation capacity and ozone formation using the model. However, determining the precise mechanisms remains challenging based on current observational data. To avoid overemphasizing marine sources, we have revised the title to "**Nitrous Acid Budgets in Coastal Atmosphere: Potential Daytime Marine Sources**" as suggested by the reviewer. The updated title emphasizes our discovery of the presence of marine sources at this site, with the specific mechanisms requiring further research in the future.

*2.    The accuracy of the model simulations is the key to the discussion in this paper. How can the accuracy of the model simulation be determined in the scenario where there are clearly 2 peaks in HONO during the observation period, but the model can only explain one of them.*

**Response:** Thanks for the reviewer's constructive and insightful comments. Upon careful examination, we have found that the late afternoon HONO peak around 18:00 is primarily attributed to specific days with relatively high primary emissions. These emissions are notably associated with International Workers' Day (May 1, 2, and 4), a residential fire incident (May 8), and dust episodes (April 27 and 28). During holiday

periods, Mount Lao experiences a significant increase in tourist numbers, coinciding with afternoon spikes in NO levels (Figure R1). Additionally, it is worth noting that transport-related influences likely contribute to afternoon peaks during dust episodes. When excluding the data from the above-selected afternoon peak days, only a noon peak of HONO is observed during the overall observation period, consistent with previous observations in clean areas (Jiang et al., 2022). Importantly, in the "sea case" we selected, there were no significant afternoon peaks in HONO concentrations (Figure 5). In the revised manuscript, we provided further clarification and included a figure illustrating the diurnal variations of HONO and other pollutants after excluding data from days affected by direct emissions in the supporting information (Figure R2). We appreciate the reviewer's valuable comments, which has substantially enhanced the reliability and quality of this paper.

[Figure]

Figure R1. Average diurnal variations of HONO and NO during the International Workers' Day. The shaded area indicates half of the standard deviation.

[Figure]

Figure R2. Comparison of diurnal variations of observed parameters in selected days with the afternoon peaks (April 27, 28, May 1, 2, 4, and 8) and overall days excluding the selected afternoon peak days.

Line 272: It is important to note that both HONO and NO$_x$ exhibit a second daytime peak in the late afternoon, which is primarily caused by specific days with primary

emissions or transport events. After excluding these days, the diurnal variation of HONO aligns more closely with observations in clean regions (Jiang et al., 2022). We have included the diurnal variation of HONO after removing days with afternoon peaks in the supporting information. Since their impact is largely confined to specific days, it does not significantly affect our subsequent analysis, particularly the "sea case" analysis.

*3. The authors only set the HONO concentration to zero in the "without HONO" scenario, which in fact still has the effect of HONO in the model. I think that the HONO-related reactions should be turned off instead of just setting HONO concentration to zero.*

**Response:** Thanks for the reviewer's insightful suggestions. In the revised manuscript, we simulated the "without HONO" scenario by setting the HONO concentration to zero and turning off the HONO-related reactions. Since we used 5-minute temporal resolution input constraints in the previous manuscript, the impact of the change is relatively minor. The modified results are as follows:

Line 425: In contrast, in the "without HONO" scenario, we turn off seven HONO production pathways summarized in Table 1 and set the input HONO concentrations to zero.

Line 435: Specifically, the absence of HONO resulted in a decrease in the net $O_3$ production rate and OH radical primary production rate in the "overall case" from 7.39 ppbv $h^{-1}$ and $1.44 \times 10^7$ molecules $cm^{-3}$ $s^{-1}$ to 3.41 ppbv $h^{-1}$ (a 54% reduction) and $2.81 \times 10^6$ molecules $cm^{-3}$ $s^{-1}$ (an 81% reduction), respectively. Regarding concentration, the absence of HONO chemistry resulted in a reduction in the average OH radical concentration from $3.6 \times 10^6$ molecules $cm^{-3}$ to $1.9 \times 10^6$ molecules $cm^{-3}$, and the peak OH concentration from $5.2 \times 10^6$ molecules $cm^{-3}$ to $2.7 \times 10^6$ molecules $cm^{-3}$. Similarly, in the marine air masses, the production rates of $O_3$ and OH decreased from 6.22 ppbv $h^{-1}$ and $7.69 \times 10^6$ molecules $cm^{-3}$ $s^{-1}$ to 3.33 ppbv $h^{-1}$ (a 46% reduction) and $2.14 \times 10^6$ molecules $cm^{-3}$ $s^{-1}$ (a 72% reduction), respectively without HONO chemistry.

Line 449: Specifically, missing marine HONO sources contributed 36% to the peak net $O_3$ production rate(from 9.24 ppbv $h^{-1}$ to 5.90 ppbv $h^{-1}$)and 28% to peak OH concentration (from $3.4 \times 10^6$ molecules $cm^{-3}$ to $2.4 \times 10^6$ molecules $cm^{-3}$).

*Minor comments:*
*1. Additional studies on marine source need to be added in the introduction section.*

**Response:** Thanks for the suggestion. We have added additional studies on marine HONO sources into the introduction section of the revised manuscript.

Line 71: The marine boundary layer (MBL) with a large air/water interface, where the ocean and atmosphere exchange trace gases, heat, and aerosol particles (Wurl et al., 2016), and the interfacial photochemistry processes often occur (Bruggemann et al., 2018), is utterly different from inland environments. The opposite diurnal variations of HONO with a peak concentration at noon at marine sites implied the different predominant HONO processes compared with polluted inland areas (Jiang et al., 2022). Furthermore, recent observations of HONO in coastal and marine regions indicate the

existence of marine HONO sources (Jiang et al., 2022; Crilley et al., 2021; Ye et al., 2016a; Yang et al., 2021a). The observed accelerated $NO_2$-to-HONO conversion in marine air masses suggests that air-marine interactions enhance HONO production (Zha et al., 2014; Yang et al., 2021a). However, the heterogeneous conversion of $NO_2$ on vast air/water interface, a potential source of marine HONO, remains uncertain (Wojtal et al., 2011; Zhu et al., 2022; Yu et al., 2021). Crilley et al. (2021) obtained a factor of 5 lower ocean-surface $NO_2$-to-HONO conversion than previous studies; there was still a debate on the importance of ocean-surface-mediated conversion of $NO_2$ into HONO. Nitrate Photolysis is believed to contribute to marine HONO sources (Ye et al., 2016a; Andersen et al., 2023), but significant controversy persists (Romer et al., 2018; Shi et al., 2021). The specific influencing factors remain unclear (Zhang et al., 2020; Andersen et al., 2023), with some studies suggesting other factors may be responsible (Yang et al., 2021a; Wojtal et al., 2011). Accordingly, Jiang et al. (2023) highlighted the contribution of the dust-surface-photocatalytic conversion of reactive nitrogen compounds to HONO formation and the important role of halogen chemistry in HONO simulation in CVAO. However, most existing studies still rely on steady-state analysis, and there is a lack of quantitative research determining if current HONO mechanisms can adequately explain observed marine daytime HONO concentrations.

*2. What are the definition criteria for daytime time ranges (7:00-17:00) in this study? The light intensity is already so low at 18:00 that the photolysis reaction has stopped?*

**Response:** Thanks for the reviewer's comment. Our choice of the daytime range from 7:00–17:00 is primarily based on two considerations. Although photolysis reactions continue at 18:00, the light intensity is relatively low, approximately one order of magnitude lower than at noon (Figure R3). Given our focus on investigating the missing daytime HONO sources, we restrict our analysis to 7:00–17:00 to minimize potential simulation errors associated with lower light intensity. Additionally, the choice of the time range is to synchronize with other observational data. Specifically, the filter sample collection started at 7:00, while carbonyl samples ended at 17:00. This ensures the consistency of our data. Overall, this choice would not alter the major conclusions of this study.

[Figure]

Figure R1. Average diurnal variation of $JNO_2$ during the observation period. The shaded area indicates the range of the standard deviation.

*3. Line 167: How are the contribution of vehicle emissions to HONO adjusted based on the environment and traffic density?*

**Response:** Thanks for the reviewer's comment. We apologize for the misleading description in the previous manuscript. In this study, we did not adjust HONO emissions based on the environment and traffic density. A commonly used HONO/NO$_x$ ratio of 0.8% (Czader et al., 2012; Lee et al., 2016; Xue et al., 2020) was employed for modeling scenarios, and sensitivity simulations were conducted using ratios of 0.4% and 1.6% to assess the impact of parameter selection uncertainty on the results. From the results shown in Figures S2, S3, and S4, the simulated HONO concentrations in all model cases remained unchanged with changes in HONO/NO$_x$. We have made the following modifications to the manuscript:

Line 185: In this study, we employed the widely used ratio of 0.8% for modeling scenarios and sensitivity simulations using ratios of 0.4% and 1.6%.

*4. Line212:The dry deposition of the other components is not considered?*

**Response:** Thanks for the reviewer's comment. We also considered the dry deposition of other components, including ozone, peroxides, carbonyls, and organic acids, in the OBM model, as described in our previous study (Xue et al., 2014). We have included this clarification in the revised manuscript.

Line 231: The dry deposition of HONO, ozone, and other species, including peroxides, carbonyls, and organic acids, are also considered in the OBM model (Xue et al., 2014).

*5. Can the authors explain the reason for the second HONO peak in the late afternoon?*

**Response:** Thanks for the reviewer's constructive and insightful comments. As explained above, the HONO peak in the late afternoon, around 18:00, is primarily due to direct vehicle emissions during International Workers' Day (May 1, 2, and 4) and a fire incident on May 8. During holiday periods, Mount Lao experiences a notable surge in tourist numbers. We also observed afternoon spikes at NO levels during these times. It is worth noting that there are also noticeable afternoon peaks during dust episodes (April 27 and 28), which may be due to transport-related influences. When excluding the data from the above-selected afternoon peak days, only a noon peak of HONO is observed during the overall observation period, consistent with previous observations in clean areas (Jiang et al., 2022). We have added clarification in the revised manuscript and included a figure depicting the diurnal variations of HONO and other pollutants after removing the data from days affected by direct emissions in the supporting information. We thank the reviewer again for the comments to improve our paper's reliability and quality.

*6. How the VOC data is used in the model, since the VOC measurements have such a low time resolution?*

**Response:** Thanks for the reviewer's comment. In this study, the daytime VOCs were

linearly interpolated to a time resolution of one hour for model constraints, whereas the nighttime VOCs were calculated according to their linear regressions with other species. Specifically, the nighttime VOCs were interpolated to one-hour time resolution using their linear regression with CO concentration. We have made the following revisions to the manuscript:

Line 159: The observed data of HONO, $O_3$, NO, $NO_2$, $SO_2$, CO, VOCs, $pNO_3^-$, Sa, temperature, RH, and pressure were averaged or interpolated to a time resolution of 5 minutes, except for VOCs and $pNO_3^-$, which were linearly interpolated to a time resolution of 1 hour to constrain the model (Yang et al., 2018).

**Reference:**

Bruggemann, M., Hayeck, N., and George, C.: Interfacial photochemistry at the ocean surface is a global source of organic vapors and aerosols, Nat Commun, 9, 2101, 10.1038/s41467-018-04528-7, 2018.

Crilley, L. R., Kramer, L. J., Pope, F. D., Reed, C., Lee, J. D., Carpenter, L. J., Hollis, L. D. J., Ball, S. M., and Bloss, W. J.: Is the ocean surface a source of nitrous acid (HONO) in the marine boundary layer?, Atmos. Chem. Phys., 21, 18213-18225, 10.5194/acp-21-18213-2021, 2021.

Czader, B. H., Rappenglück, B., Percell, P., Byun, D. W., Ngan, F., and Kim, S.: Modeling nitrous acid and its impact on ozone and hydroxyl radical during the Texas Air Quality Study 2006, Atmos. Chem. Phys., 12, 6939-6951, 10.5194/acp-12-6939-2012, 2012.

Jiang, Y., Xue, L., Shen, H., Dong, C., Xiao, Z., and Wang, W.: Dominant Processes of HONO Derived from Multiple Field Observations in Contrasting Environments, Environ. Sci. Technol. Lett., 9, 258-264, 10.1021/acs.estlett.2c00004, 2022.

Jiang, Y., Hoffmann, E. H., Tilgner, A., Aiyuk, M. B. E., Andersen, S. T., Wen, L., van Pinxteren, M., Shen, H., Xue, L., Wang, W., and Herrmann, H.: Insights into NOx and HONO chemistry in the tropical marine boundary layer at Cape Verde during the MarParCloud campaign, Journal of Geophysical Research: Atmospheres, 10.1029/2023jd038865, 2023.

Lee, J. D., Whalley, L. K., Heard, D. E., Stone, D., Dunmore, R. E., Hamilton, J. F., Young, D. E., Allan, J. D., Laufs, S., and Kleffmann, J.: Detailed budget analysis of HONO in central London reveals a missing daytime source, Atmos. Chem. Phys., 16, 2747-2764, 10.5194/acp-16-2747-2016, 2016.

Wurl, O., Stolle, C., Van Thuoc, C., The Thu, P., and Mari, X.: Biofilm-like properties of the sea surface and predicted effects on air–sea $CO_2$ exchange, Prog. Oceanogr., 144, 15-24, 10.1016/j.pocean.2016.03.002, 2016.

Xue, C., Zhang, C., Ye, C., Liu, P., Catoire, V., Krysztofiak, G., Chen, H., Ren, Y., Zhao, X., Wang, J., Zhang, F., Zhang, C., Zhang, J., An, J., Wang, T., Chen, J., Kleffmann, J., Mellouki, A., and Mu, Y.: HONO Budget and Its Role in Nitrate Formation in the Rural North China Plain, Environ Sci Technol, 54, 11048-11057, 10.1021/acs.est.0c01832, 2020.

Xue, L. K., Wang, T., Gao, J., Ding, A. J., Zhou, X. H., Blake, D. R., Wang, X. F., Saunders, S. M., Fan, S. J., Zuo, H. C., Zhang, Q. Z., and Wang, W. X.: Ground-level ozone in four Chinese cities: precursors, regional transport and heterogeneous processes, Atmos. Chem. Phys., 14, 13175-13188, 10.5194/acp-14-13175-2014, 2014.

Yang, X., Xue, L., Wang, T., Wang, X., Gao, J., Lee, S., Blake, D. R., Chai, F., and Wang, W.: Observations and Explicit Modeling of Summertime Carbonyl Formation in Beijing:

Identification of Key Precursor Species and Their Impact on Atmospheric Oxidation Chemistry, Journal of Geophysical Research: Atmospheres, 123, 1426-1440, 10.1002/2017jd027403, 2018.

---

## Author Comment (AC2)

**Response to Editors and Reviewers**
We gratefully thank the reviewers for their constructive comments and suggestions to improve the manuscript. As detailed below, the reviewers' comments are shown in *black italic*; our response to the comments is in blue. New or modified text is in red.

**Referee 2:**

*General comments:*

*HONO is an important source of OH radicals in the atmosphere. Elucidating the characteristics and formation mechanisms of HONO is vital to understand the OH budget of OH. By combining modeling and field studies, Zhong et al provide evidence of a significant unidentified daytime marine source of HONO. Further, this missing HONO source is likely photochemical induced. This work has important implications for atmospheric chemistry in coastal and marine areas and will motivate further work on this topic in due course. This manuscript is well written and ACP is an appropriate venue. I would recommend the paper for publication after the following issues are addressed.*

**Response:** Thanks for the reviewer's comments and suggestions. We have addressed the specific comments and revised the manuscript accordingly. For clarity, the reviewer's comments are listed below in *black italic*, while our responses and changes in the manuscript are shown in blue and red, respectively.

*Specific comments:*

*1.   Line 1: Should it be "the presence of a daytime marine source"?*

**Response:** Thanks for the reviewer's suggestion. According to the suggestions of reviewer 1 and reviewer 2, we have revised the title to "**Nitrous Acid Budgets in Coastal Atmosphere: Potential Daytime Marine Sources**" to emphasize our discovery of the presence of marine sources in the coastal atmosphere.

*2.   Line 53: It would be helpful to introduce the missing source of daytime HONO.*

**Response:** Thanks for the reviewer's suggestion. We have added the introduction about the missing source of daytime HONO in the revised manuscript.

Line 52: Despite its short daytime atmospheric lifetime, HONO is frequently observed at high concentrations at noon (Ye et al., 2016; Yang et al., 2021; Jiang et al., 2023). Traditional mechanisms cannot fully explain these observed daytime HONO peaks, indicating the presence of additional daytime HONO sources (missing sources of daytime HONO). Over recent decades, researchers have extensively investigated the missing sources of daytime HONO in various environments (Kleffmann, 2007; Lee et al., 2016; Jiang et al., 2022; Zhang et al., 2022).

*3.   Lines 137-138: Would the data averaging procedure introduce uncertainties to the subsequent analysis considering that different period of time was chosen for different days?*

**Response:** Thanks for the reviewer's comment. To confirm the reliability of our

conclusions, we conducted simulation comparisons using a case study on May 10, which was primarily influenced by marine air masses for almost an entire day. The simulation results show that even with the updated model, the simulated daytime HONO concentration is still significantly lower than the observed value (0.09 ppbv versus 0.29 ppbv). The missing HONO production rate is up to 0.51 ppbv h$^{-1}$, similar to the average case. Therefore, the data averaging procedure does not affect our conclusions. However, as the reviewer noted, averaging with limited data introduces uncertainties. Future research should continue with a larger dataset.

[Figure]

Figure R1. Comparison of observed and modeled daytime (7:00–17:00) HONO concentrations and modeled HONO budgets on May 10th, primarily influenced by marine air masses for almost an entire day.

4.  *Lines 152: What are the unconstrained species?*

**Response:** Thanks for the reviewer's comment. The unconstrained species refers to species included in the MCM model (over 6900 species) that were not constrained with input from observed data. These primarily include atmospheric radicals and some secondary VOCs.

5.  *Lines 161-164: What is the HONO/NOx value used in this study?*

**Response:** Thanks for the reviewer's comment. In this study, a commonly used HONO/NO$_x$ ratio of 0.8% (Czader et al., 2012; Lee et al., 2016; Xue et al., 2020) was employed for modeling scenarios, and sensitivity simulations were conducted using ratios of 0.4% and 1.6% to assess the impact of parameter selection uncertainty on the results. We have made the following modifications to the manuscript:

Line 185: In this study, we employed the widely used ratio of 0.8% for modeling scenarios and sensitivity simulations using ratios of 0.4% and 1.6%.

6.  *Lines 250-252: There is another peak of HONO at around 18:00. What are the potential reasons?*

**Response:** Thanks for the reviewer's constructive and insightful comments. Upon careful examination, we have found that the late afternoon HONO peak around 18:00 is primarily attributed to specific days with relatively high primary emissions. These

emissions are notably associated with International Workers' Day (May 1, 2, and 4), a residential fire incident on May 8, and dust episodes on April 27 and 28. During holiday periods, Mount Lao experiences a significant increase in tourist numbers, coinciding with afternoon spikes in NO levels (Figure R2). Additionally, it is worth noting that transport-related influences likely contribute to afternoon peaks during dust episodes. When excluding the data from the above-selected afternoon peak days, only a noon peak of HONO is observed during the overall observation period, consistent with previous observations in clean areas (Jiang et al., 2022). Importantly, in the "sea case" we selected, there were no significant afternoon peaks in HONO concentrations (Figure 5). In the revised manuscript, we provided further clarification and included a figure illustrating the diurnal variations of HONO and other pollutants after excluding data from days affected by direct emissions in the supporting information (Figure R3). We appreciate the reviewer's valuable comment, which has substantially enhanced the reliability and quality of our paper.

[Figure]

Figure R2. Average diurnal variations of HONO and NO during the International Workers' Day. The shaded area indicates half of the standard deviation.

[Figure]

Figure R3. Comparison of diurnal variations of observed parameters in days with the selected afternoon peaks (April 27, 28, May 1, 2, 4, and 8) and overall days excluding the selected afternoon peak days. We have also included this figure in the supporting information.

*7. Lines 268-270: I am curious about the performance of the updated OBM on non-polluted periods. Do the model results of HONO agree well with the observations?*

**Response:** Thanks for the reviewer's constructive comment and suggestion. Similar to photochemical and dust episodes, the updated model with revised parameters performs well in simulating daytime HONO budgets during non-polluted periods (excluding the influence of marine air masses). Figure R4 compares the observed and simulated daytime (7:00–17:00) HONO concentrations during the non-polluted period. The simulated average HONO concentration is 0.59 ppbv, close to the observed value (0.67 ppbv). The index of agreement (IOA) value during the non-polluted period is 0.79, indicating that the updated OBM effectively reproduces the observation of HONO. This result suggests that the model performs relatively well in simulating non-marine sources at the observation site.

[Figure]

Figure R4. Comparison of observed and modeled daytime (7:00–17:00) HONO concentrations in the non-polluted period.

*8. Lines 340-342: Though the authors focus on the sources of HONO during the daytime, the nighttime observations may constrain the non-photochemical sources of HONO. Could the model capture the high nighttime HONO/NO$_2$ ratio in the "sea case"?*

**Response:** Thanks for the reviewer's comment. We observed high HONO/NO$_2$ ratios during the nighttime in the "sea case" (0.12 ± 0.11), which suggests potentially significant roles of the heterogeneous NO$_2$ conversion or other sources without the involvement of NO$_2$. After incorporating updated mechanisms into our model, the simulated HONO concentrations significantly increased and maintained relatively high nighttime HONO levels. However, the simulated values still fell below the observed values (Figure R5). The simulated HONO/NO$_x$ ratio reached 0.07 ± 0.01, lower than the observed value. This indicates that there may be other sources of HONO during nighttime, such as microbial activity or soil emissions (Oswald et al., 2013; Song et al., 2022), or potentially higher efficiency in the conversion of NO$_2$ in marine air masses (Yang et al., 2021; Yabushita et al., 2009). However, compared to daytime, the nighttime simulations showed relatively better agreement with observations, and daytime HONO has a more significant impact on atmospheric oxidation and ozone

formation. Therefore, in this paper, we primarily focused our discussion on daytime HONO.

[Figure]

Figure R5. Comparison of the observed and modeled HONO concentrations in the "sea case".

*9. Lines 374-376: The interfacial chemistry may lead to a high uptake coefficient of $NO_2$ on aerosol. Previous laboratory studies have found that the uptake coefficient of $NO_2$ on the aerosol surface can reach $2 \times 10^{-4}$ (see Liu and Abbatt (2021); Yabushita et al. (2009) and references therein).*

**Response:** Thanks for the reviewer's comments and recommended references. The measured $NO_2$ uptake coefficient of $2 \times 10^{-4}$ by Liu and Abbatt (2021) represents a relatively high value observed in laboratory studies. However, upon carefully comparing their experimental conditions with our field observations, we find it challenging to achieve such values under field conditions. Firstly, their laboratory experiments involved a high $SO_2$ concentration of 280 ppbv, which can enhance $NO_2$ uptake, whereas the $SO_2$ level during our observation was around 1 ppbv. Furthermore, Liu and Abbatt (2021) reported that at pH levels below 4, the pathway for $NO_2$ uptake becomes negligible, suggesting that the $NO_2$ uptake coefficients should be considerably lower under these conditions. Our thermodynamic model estimated an aerosol pH of 3.1 during the observation. Therefore, we consider it challenging to achieve $NO_2$ uptake coefficients of this magnitude during our observations, and even the value of $2 \times 10^{-4}$ is still twice less than the required $NO_2$ uptake coefficients. For the study of Yabushita et al. (2009), they measured initial-state (1 ms) uptake coefficients, and the equilibrium-state $NO_2$ uptake coefficients would likely be considerably lower (Yu et al., 2021). However, it should be noted that their study mentioned that the presence of halogens could enhance $NO_2$ uptake, which may contribute to the relatively high conversion of $NO_2$ to HONO in marine environments. We have cited and discussed these two papers in our manuscript. Once again, we thank the reviewer's comments.

Line 401: Although $NO_2$ uptake coefficients on the order of $10^{-4}$ have been measured in laboratory experiments under conditions of high $SO_2$ concentration (280 ppbv) and moderate acidity (pH = 5) (Liu and Abbatt, 2021), our observational site features lower

SO$_2$ concentrations (~ 1ppbv) and slightly acidic aerosols (pH = 3.1). These conditions suggest that the uptake coefficients should be considerably lower than the laboratory-measured 2×10$^{-4}$. It is worth noting that previous research has indicated that the presence of halogens can enhance NO$_2$ uptake, which could potentially explain the higher NO$_2$ to HONO conversion ratios in marine environments (Yabushita et al., 2009). However, further research is needed to explore this possibility. Overall, the observed missing HONO source in the "sea case" cannot be explained by the current photochemical processes.

*10. Figure 4: Do the blue lines in Figure 4c, d represent the observed HONO production rate?*

**Response:** The blue lines in Figures 4c and 4d represent the observed HONO production rate. We have modified the legends of Figures 4 and 6 by changing "HONO$_{obs}$" to "P(HONO)$_{obs}$" accordingly.

[Figure]

**Figure 4.** Daytime HONO budgets in dust (a, c) and (b, d) photochemical period at Mount Lao. The base case only considered the homogeneous reaction of NO + OH, and the model case considered the updated HONO sources described in this study.

[Figure]

**Figure 6.** Comparison of the observed and modeled daytime (7:00–17:00) HONO concentrations and modeled HONO budgets in the "land case" (a, c) and the "sea case" (b, d).

***Technical comments:***

1. *Line 116: Give the full words of VOC in the first appearance.*

**Response:** We have corrected it accordingly.

Line 130: During the field campaign, fifty-seven VOC (volatile organic compound) canister samples were collected at 2-hour intervals from 9:00–19:00 local time on pollution episode days and at 6-hour intervals from 9:00–21:00 on non-episode days.

***Reference:***

Czader, B. H., Rappenglück, B., Percell, P., Byun, D. W., Ngan, F., and Kim, S.: Modeling nitrous acid and its impact on ozone and hydroxyl radical during the Texas Air Quality Study 2006, Atmos. Chem. Phys., 12, 6939-6951, 10.5194/acp-12-6939-2012, 2012.

Jiang, Y., Xue, L., Shen, H., Dong, C., Xiao, Z., and Wang, W.: Dominant Processes of HONO Derived from Multiple Field Observations in Contrasting Environments, Environ. Sci. Technol. Lett., 9, 258-264, 10.1021/acs.estlett.2c00004, 2022.

Jiang, Y., Hoffmann, E. H., Tilgner, A., Aiyuk, M. B. E., Andersen, S. T., Wen, L., van Pinxteren, M., Shen, H., Xue, L., Wang, W., and Herrmann, H.: Insights into NOx and HONO chemistry in the tropical marine boundary layer at Cape Verde during the MarParCloud campaign, Journal of Geophysical Research: Atmospheres, 10.1029/2023jd038865, 2023.

Kleffmann, J.: Daytime sources of nitrous acid (HONO) in the atmospheric boundary layer, Chemphyschem, 8, 1137-1144, 10.1002/cphc.200700016, 2007.

Lee, J. D., Whalley, L. K., Heard, D. E., Stone, D., Dunmore, R. E., Hamilton, J. F., Young, D. E., Allan, J. D., Laufs, S., and Kleffmann, J.: Detailed budget analysis of HONO in central London reveals

a missing daytime source, Atmos. Chem. Phys., 16, 2747-2764, 10.5194/acp-16-2747-2016, 2016.

Liu, T. and Abbatt, J. P. D.: Oxidation of sulfur dioxide by nitrogen dioxide accelerated at the interface of deliquesced aerosol particles, Nat Chem, 13, 1173-1177, 10.1038/s41557-021-00777-0, 2021.

Oswald, R., Behrendt, T., Ermel, M., Wu, D., Su, H., Cheng, Y., Breuninger, C., Moravek, A., Mougin, E., and Delon, C.: HONO emissions from soil bacteria as a major source of atmospheric reactive nitrogen, Science, 341, 1233-1235, 2013.

Song, W., Liu, X. Y., Houlton, B. Z., and Liu, C. Q.: Isotopic constraints confirm the significant role of microbial nitrogen oxides emissions from the land and ocean environment, Natl Sci Rev, 9, nwac106, 10.1093/nsr/nwac106, 2022.

Xue, C., Zhang, C., Ye, C., Liu, P., Catoire, V., Krysztofiak, G., Chen, H., Ren, Y., Zhao, X., Wang, J., Zhang, F., Zhang, C., Zhang, J., An, J., Wang, T., Chen, J., Kleffmann, J., Mellouki, A., and Mu, Y.: HONO Budget and Its Role in Nitrate Formation in the Rural North China Plain, Environ Sci Technol, 54, 11048-11057, 10.1021/acs.est.0c01832, 2020.

Yabushita, A., Enami, S., Sakamoto, Y., Kawasaki, M., Hoffmann, M., and Colussi, A.: Anion-catalyzed dissolution of NO2 on aqueous microdroplets, The Journal of Physical Chemistry A, 113, 4844-4848, 2009.

Yang, J., Shen, H., Guo, M.-Z., Zhao, M., Jiang, Y., Chen, T., Liu, Y., Li, H., Zhu, Y., Meng, H., Wang, W., and Xue, L.: Strong marine-derived nitrous acid (HONO) production observed in the coastal atmosphere of northern China, Atmos. Environ., 244, 10.1016/j.atmosenv.2020.117948, 2021.

Ye, C., Zhou, X., Pu, D., Stutz, J., Festa, J., Spolaor, M., Tsai, C., Cantrell, C., Mauldin, R. L., 3rd, Campos, T., Weinheimer, A., Hornbrook, R. S., Apel, E. C., Guenther, A., Kaser, L., Yuan, B., Karl, T., Haggerty, J., Hall, S., Ullmann, K., Smith, J. N., Ortega, J., and Knote, C.: Rapid cycling of reactive nitrogen in the marine boundary layer, Nature, 532, 489-491, 10.1038/nature17195, 2016.

Yu, C., Wang, Z., Ma, Q., Xue, L., George, C., and Wang, T.: Measurement of heterogeneous uptake of NO2 on inorganic particles, sea water and urban grime, J Environ Sci (China), 106, 124-135, 10.1016/j.jes.2021.01.018, 2021.

Zhang, X., Tong, S., Jia, C., Zhang, W., Li, J., Wang, W., Sun, Y., Wang, X., Wang, L., Ji, D., Wang, L., Zhao, P., Tang, G., Xin, J., Li, A., and Ge, M.: The Levels and Sources of Nitrous Acid (HONO) in Winter of Beijing and Sanmenxia, Journal of Geophysical Research: Atmospheres, 127, 10.1029/2021jd036278, 2022.